# ENHANCING TEMPORAL KNOWLEDGE GRAPH COMPLETION WITH GLOBAL SIMILARITY AND WEIGHTED SAMPLING

## ABSTRACT

Temporal Knowledge Graph (TKG) completion models traditionally assume access to the entire graph during training. This overlooks challenges stemming from the evolving nature of TKGs, such as: (i) the model's requirement to generalize and assimilate new knowledge, and (ii) the task of managing new or unseen entities that often have sparse connections. In this paper, we present an incremental training framework specifically designed for TKGs, aiming to address entities that are either not observed during training or have sparse connections. Our approach combines a model-agnostic enhancement layer with a weighted sampling strategy, that can be augmented to and improve any existing TKG completion method. The enhancement layer leverages a broader, global definition of entity similarity, moving beyond mere local neighborhood proximity of GNN-based methods. Our evaluations, conducted on two benchmark datasets, demonstrate that our framework outperforms existing methods in overall link prediction, inductive link prediction, and in addressing long-tail entities. Notably, our method achieves a 10% improvement in MRR for one dataset and a 15% boost for another. The results underscore the potential of our approach in mitigating catastrophic forgetting and enhancing the robustness of TKG completion methods, especially in an incremental training context.

## 1 INTRODUCTION

Knowledge graphs have become a fundamental tool for studying the underlying structure of multi-relational data in the real world  Liang et al. (2022).These graphs encapsulate factual information as a set of triplets, each consisting of a subject entity, a relation, and an object entity. This facilitates the analysis of complex relations and interactions within the data, establishing KGs as essential components that can enhance various other NLP tasks such as question answering, information retrieval, and large language models Pan et al. (2023). However, KGs often grapple with a problem of incompleteness. Despite advancements in extraction methods, the occurrence of missing facts remains prevalent, significantly impacting downstream applications. As a result, the task of knowledge graph completion, i.e. predicting these missing facts has become of paramount importance Wang et al. (2022); Huang et al. (2022); Shen et al. (2022).

Most real-world Knowledge graphs are often derived from dynamic data streams, such as evolving news sources. As these streams introduce new entities and relationships, temporal information becomes integral to KG facts. In semantic KGs like Yago Kasneci et al. (2009), facts are associated with time intervals, e.g., *(Obama, President, United States, 2009-2017)*. In contrast, Temporal event-centric knowledge graphs (TKGs) like ICEWS Boschee et al. (2015) link facts to specific timestamps, capturing precise interaction moments. Thus, *(Obama, meet, Merkel)* in a TKG might recur multiple times between 2009 and 2017, reflecting distinct events. This distinction underscores the pronounced dynamism of event-centric TKGs compared to semantic KGs.

Temporal knowledge graph (TKG) completion has recently emerged as a prominent research area, leading to the development of methods that derive temporal representations from historical interactions Jin et al. (2019); Zhu et al. (2020); Sun et al. (2021); Niu & Li (2023). However, many TKG completion techniques assume access to all facts during training. This assumption overlooks two critical challenges: (i) the need to integrate new information from fresh facts while retaining prior

knowledge, and (ii) the evolving distribution of entities, characterized by the emergence of new entities or entities that appear sporadically over specific periods.

Traditional approaches typically initialize an embedding table for all entities, which is then learned during training. Yet, for entities unseen during training, their lack of connections in the graph hinder the update of their embeddings, rendering them effectively untrained. While Graph Neural Network (GNN) based methods aim to infer embeddings for these unseen entities using their temporal neighborhoods, they often struggle when these neighborhoods are sparse. Few works evaluate their method under the inductive link prediction setup Han et al. (2020b); Sun et al. (2021), where edges during inference involve previously unseen entities. Even fewer, such as the method by Sun et al. (2021), explicitly address these entities to derive robust representations.

Moreover, while some methods can inductively generate embeddings for new entities, they often neglect to reintegrate this information into the model. As a result, both entity embeddings and model parameters remain static, not reflecting the newly acquired knowledge. Direct retraining with new facts is computationally demanding, while simple fine-tuning risks overfitting and catastrophic forgetting. Some recent studies have introduced incremental training frameworks for TKG completion, but these often preserve the base model's architecture, making them ill-suited for unseen entities unless inherently designed for such scenarios.

In this study, we present an incremental training framework tailored for Temporal Knowledge Graphs (TKGs) to effectively handle entities that are either unseen or sparsely represented in the data. We propose a model-agnostic enhancement layer, specifically designed for GNN-based TKG completion methods. This layer seeks to enhance the representations of entities with limited local connections by leveraging a broader, global definition of entity similarity, moving beyond mere local neighborhood proximity. To further refine our approach, we adopt a weighted sampling strategy during training, emphasizing edges associated with infrequently occurring entities. This strategy enhances the accuracy of link prediction, particularly for long-tail and previously unseen entities. To further enhance link prediction accuracy, we adopt a weighted sampling strategy during training that emphasizes edges associated with infrequently occurring entities, particularly benefiting long-tail and previously unseen entities. For validation, we introduce two benchmark datasets designed for incremental learning, assessing our model across various scenarios. Our evaluations focus on overall link prediction, inductive link prediction, and the model's capability to effectively address long-tail entities. A summary of the main contributions of our work are as follows:

- **Incremental Training Framework for TKGs:** We introduce a novel incremental training framework tailored for Temporal Knowledge Graphs, designed to effectively handle entities that are either unseen during training or have sparse representations in the data.
- **Model-Agnostic Enhancement Layer:** We propose a unique enhancement layer that can be integrated with various GNN-based TKG completion methods. This layer enriches entity representations by leveraging a broader, global definition of entity similarity, surpassing the traditional reliance on local neighborhood proximity.
- **Weighted Sampling Strategy:** We implement a weighted sampling strategy during the training process, emphasizing edges associated with infrequently occurring entities.
- **Benchmark Datasets for Incremental Learning:** For validation, we curate two benchmark datasets specifically designed for incremental learning, allowing for a comprehensive assessment of our model across diverse scenarios.
- **Comprehensive Evaluation:** Our evaluations provide insights into the effectiveness of our approach in various link prediction tasks, with a focus on overall link prediction, inductive link prediction, and the model's capability to address long-tail entities.

## 2 RELATED WORK

This study intersects with research on TKG completion, continual learning techniques, and the recent advancements in continual learning specifically tailored for knowledge graphs.

**Temporal Knowledge Graph Completion**. Methods for TKG completion can be primarily divided into two categories based on their approach to encoding temporal information. Translation-based approaches, such as those introduced by Leblay & Chekol (2018); García-Durán et al. (2018); Dasgupta

et al. (2018); Wang & Li (2019); Jain et al. (2020); Sadeghian et al. (2021), employ an embedding representation like a vector Leblay & Chekol (2018); Jain et al. (2020) or a hyperplane Dasgupta et al. (2018); Wang & Li (2019) to encode event timestamps and establish a function to transition an initial embedding to one that is time-aware. Others model the temporal information using shallow encoders Xu et al. (2019); Han et al. (2020a) or by employing sequential neural architectures Trivedi et al. (2017); Jin et al. (2020); Wu et al. (2020); Zhu et al. (2020); Han et al. (2020b;c); Li et al. (2021); Zhang et al. (2023). For instance, DyERNIE Han et al. (2020a) introduces a non-Euclidean embedding method in hyperbolic space. Trivedi et al. (2017) represents events using point processes, while Jin et al. (2020); Wu et al. (2020) employs a recurrent structure to aggregate past timestamp entity neighborhoods. A few works Han et al. (2020b); Sun et al. (2021); Han et al. (2021a) evaluate their performance on inductive link prediction. None of the above works consider a growing TKG and thus cannot continually learn knowledge.

**Continual Learning**. Continual learning, also known as lifelong learning, is a paradigm where a stream of task is introduced to and learned by the model sequentially. A primary challenge in this paradigm is catastrophic forgetting, where learning new tasks in the sequence degrades the model's performance on previously learned tasks. Experience replay Li & Hoiem (2018) is a prominent strategy to counteract forgetting, where past task samples are replayed during model updates to retain previously acquired knowledge. To manage a fixed-size memory buffer, it becomes essential to judiciously select and discard representative samples. Schaul et al. (2016) suggests selecting samples that have the most significant impact on the loss function from past tasks.

To obviate the need for a memory buffer, generative models can be employed to produce pseudo-samples. For exmaple, Shin et al. (2017) leverages adversarial learning for this purpose. Another strategy, weight consolidation Zenke et al. (2017); Kirkpatrick et al. (2017), identifies and consolidates crucial weights that encode knowledge from past tasks. Consequently, new tasks primarily utilize the remaining learnable weights. Our proposed framework integrates both strategies to optimize performance.

**Continual Graph Learning**. The application of continual learning to graph structures is a relatively recent development. A handful of studies have addressed dynamic heterogeneous networks Tang & Matteson (2021); Wang et al. (2020); Zhou & Cao (2021) and semantic knowledge graphs Song & Park (2018); Daruna et al. (2021); Wu et al. (2021a). Specifically, Song & Park (2018); Daruna et al. (2021) integrate class incremental learning models with static translation-based methods, such as TransE Bordes et al. (2013), to address the continual KG embedding challenge. TIE Wu et al. (2021a) primarily focuses on semantic KGs and generates yearly graph snapshots by converting a fact with a time interval into multiple timestamped facts. This conversion can result in the loss of granular temporal details, which are essential for temporal event-centric knowledge graphs. Mirtaheri et al. (2023) introduces a framework for TKG completion based on regularization and experience replay. However, their approach does not address the challenge posed by long-tail entities.

## 3  PROBLEM FORMULATION

In this section, we provide formal definitions for continual temporal knowledge graph completion.

### 3.1  TEMPORAL KNOWLEDGE GRAPH COMPLETION

A Temporal Knowledge Graph (TKG) $G = \langle \mathcal{Q}, \mathcal{E}, \mathcal{R} \rangle$ encapsulates a series of events as a set of quadruples $\mathcal{Q} = \{(s, r, o, \tau) | s, o \in \mathcal{E}, r \in \mathcal{R}\}$. Here, $\mathcal{E}$ and $\mathcal{R}$ are the sets of entities and relations respectively, and $\tau$ denotes the timestamp of the event's occurrence. These events signify one-time interactions between entities at a specific point in time. The objective of temporal knowledge graph completion is to predict potential interactions between two entities at a given time. This prediction can be accomplished by either predicting the object entity, given the subject and relation at a certain time, or by predicting the relation between entities, given the subject and object at a specific time. For a given Temporal Knowledge Graph with observations made up to time $T$. The completion task can be performed in two setups:

- **Interpolation** in which we predict events timestamped before $T$.
- **Extrapolation** in which we predict events that are timestamped beyond $T$.

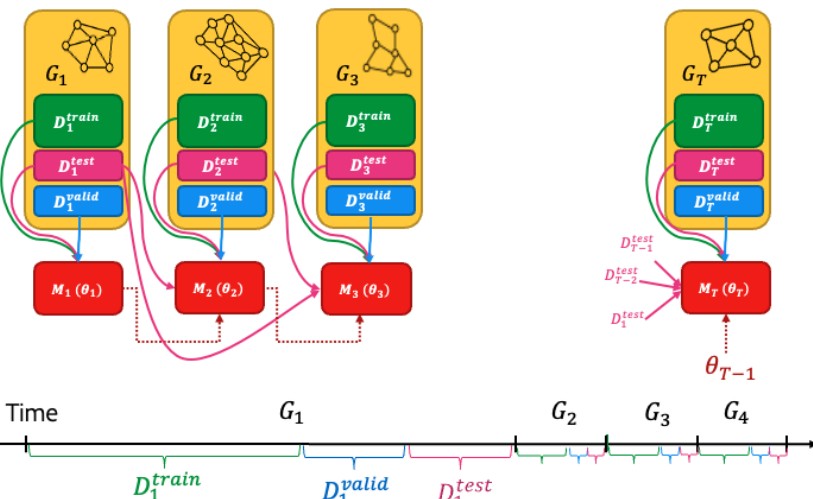

Figure 1: Continual Extrapolation Setup. The arrow depicts the time. For extrapolation, the edge timestamps in the validation and test set do not have any overlap.

In this work, we focus on extrapolation.

### 3.2 CONTINUAL LEARNING FRAMEWORK FOR TEMPORAL KNOWLEDGE GRAPHS

A Temporal Knowledge Graph $G$ can be represented as a stream of graph snapshots $\langle G_1, G_2, \ldots, G_T \rangle$ arriving over time. Each snapshot $G_t = \langle \mathcal{Q}_t, \mathcal{E}_t, \mathcal{R}_t \rangle$ includes $\mathcal{Q}_t = \{(s, r, o, \tau) | s, o \in \mathcal{E}_t, r \in \mathcal{R}_t, \tau \in [\tau_t, \tau_{t+1})\}$, which is the set of quadruples that occurred within the time interval $[\tau_t, \tau_{t+1})$, and $\mathcal{E}_t, \mathcal{R}_t$ are the sets of entities and relations at time $t$.

Incremental training of a TKG completion method involves updating the model parameters $\mathcal{M}$ as new graph snapshots, each comprising a set of events, become available over time. This process strives to retain previously learned information while assimilating new patterns. Formally, we define a set of tasks $\langle \mathcal{T}_1, \ldots, \mathcal{T}_T \rangle$, where each task $\mathcal{T}_t = \left( D_t^{train}, D_t^{test}, D_t^{val} \right)$ consists of disjoint subsets of the $G_t$ events. The trained model $\mathcal{M}$ is thus represented as a stream of models $\mathcal{M} = \langle \mathcal{M}_1, \ldots, \mathcal{M}_T \rangle$, with corresponding parameter sets $\theta = \langle \theta_1, \theta_2, ..., \theta_T \rangle$, trained incrementally as a stream of tasks $\mathcal{T} = \langle \mathcal{T}_1, \mathcal{T}_2, ..., \mathcal{T}_T \rangle$ arrives. Figure 1 depicts the incremental training and evaluation setup for temporal extrapolation.

## 4 METHODOLOGY

Temporal Knowledge Graphs (TKGs) are dynamic structures where entities and their relationships evolve over time. A significant challenge in TKG completion is the long-tailed distribution of entities, where many entities appear infrequently, leading to sparse connections. This sparsity is accentuated in growing TKGs, where entity distributions shift over time. An entity might have dense connections at one time point and sparse connections at another. This dynamic nature poses two primary challenges: (i) model parameters become biased towards densely connected entities in recent times, and (ii) representations of long-tailed entities remain outdated, not reflecting the current graph patterns.

### 4.1 MODEL-AGNOSTIC ENHANCEMENT LAYER

To address the aforementioned challenges, we propose a model-agnostic enhancement layer that extends beyond the local neighborhood of entities. This layer can be integrated with any TKG completion method. It enhances the representation of an entity by incorporating a temporal global representation from other similar entities, even if they aren't in the immediate vicinity.

The enhanced entity representation $e_s$ is formulated as:

$$e_s = \lambda f(s) + \frac{1}{1 + \exp(d_s)}(1 - \lambda)g(s) \tag{1}$$

Here, $f(s)$ denotes the embeddings of entity $s$ from the underlying TKG completion model or any intermediate layer of a GNN-based method. The function $g(s)$ is the enhancement function, which is temporal and derives a representation from a set of entities similar to $s$ at time $t$, represented as $S_t(s)$.

### 4.1.1 DEFINING ENTITY SIMILARITY

Entity similarity is crucial for the enhancement function. In this work, we define a relation-based similarity. Two entities $s_1$ and $s_2$ are deemed similar if they share a connection via the same relation. Formally, the set of similar entities at time $t$ for a given entity $s$ and relation $r$ is:

$$S_t(s, r) = \{s_i | (s_i, r, o_i, t_i) \in G, t_i < t\} \tag{2}$$

### 4.1.2 ENHANCEMENT FUNCTION

The enhancement function, given a query $(s, r, ?, t)$, is defined as:

$$g(s, r) = \frac{\sum_{s_i \in S_t(s,r)} w_i e_{s_i}}{\sum w_i}, \quad w_i = \frac{1}{1 + \exp(t - t_i)} \tag{3}$$

For computational efficiency, we consider only a subset of $S_t(s, r)$, focusing on the most recent events and their corresponding entities.

### 4.2 WEIGHTED FREQUENCY-BASED SAMPLING

In traditional training processes, there exists an inherent bias towards entities with high degrees that appear in numerous quadruples. To counteract this bias, we introduce a weighted frequency-based sampling strategy. This strategy samples quadruples from the training set based on the inverse frequency of either the subject or object entity.

Formally, the likelihood of a quadruple $(s, r, o, t)$ being selected for training, denoted by $\phi$, is given by:

$$\phi(s, r, o, t) = \alpha \mathcal{P}(s, r, o, t) + (1 - \alpha)\mathcal{U}(s, r, o, t) \tag{4}$$

Where $\mathcal{P}(s, r, o, t)$ represents the probability of sampling the quadruple $(s, r, o, t)$ and is directly proportional to the inverse frequency of entities $s$ and $o$. On the other hand, $\mathcal{U}$ denotes a uniform probability function.

To be more specific, $\mathcal{P}(s, r, o, t)$ is defined to be proportional to either:

$$\text{mean}\left(\frac{1}{\text{freq}(s)}, \frac{1}{\text{freq}(o)}\right)$$

or

$$\text{max}\left(\frac{1}{\text{freq}(s)}, \frac{1}{\text{freq}(o)}\right)$$

In the context of continual learning, the frequency of an entity is determined by the number of times it appears in quadruples up to the current training step. Notably, we do not compute the entity frequency based on the entire dataset.

## 5 EXPERIMENTS

Our experiments focus on entity prediction, commonly referred to as knowledge graph completion. Through these experiments, we aim to address the following research questions:

- **Q1**: Can our framework effectively generalize to new facts while preserving previously acquired knowledge?

| Dataset | # entities | # relations | # snapshots | # quads ($G_1$) | Avg. quads ($G_{i>1}$) |
|---------|-----------|------------|-------------|-----------------|------------------------|
| ICEWS14 | 7128 | 230 | 33 | 28k/3.7k/4k | 1k/0.3k/0.3k |
| ICEWS18 | 23039 | 230 | 16 | 244k/45k/43k | 8K/1k/1k |

Table 1: Dataset statistics

- **Q2**: How well does our approach generalize to links involving new entities and entities with sparse neighborhoods?

- **Q3**: What is the contribution of each component of our framework to specific challenges, such as incremental learning and handling long-tail entities?

The following sections will describe the datasets, evaluation setup, results, and subsequent analysis.

## 5.1 DATASET CONSTRUCTION

We use the Integrated Crisis Early Warning System (ICEWS) dataset, which records interactions among geopolitical actors with daily event timestamps. For evaluation benchmarks, we select a specific interval from the ICEWS dataset. We then generate temporal snapshots by partitioning the data into sets of quadruples with unique timestamps. The initial snapshot contains 50% of the selected interval, and the remaining data is segmented into non-overlapping periods of size $w$. Each snapshot is further divided into training, validation, and test sets based on their timestamps. In line with the extrapolation setup, the training set timestamps are earlier than those in the validation set, which are in turn earlier than the test set timestamps, as depicted in Figure 1. We utilize the following dataset configurations: ICEWS 2018 and ICEWS 2014. For ICEWS 2014, the first seven months are designated as the initial snapshot, while subsequent snapshots each span a seven-day period. Table 1 summarizes the statistics of the datasets.

## 5.2 EVALUATION SETUP

Our primary objective is to assess our model's performance under both continual and inductive learning scenarios. To our knowledge, this is the first study that investigates inductive and continual learning for temporal knowledge graphs in an extrapolation setup. Our approach aligns closely with prior works Wu et al. (2021b); Mirtaheri et al. (2023); Cui et al. (2023) that explore continual learning for node classification in heterogeneous graphs and semantic knowledge graph completion.

We partition the quadruples of the Temporal Knowledge Graph (TKG) into sequential temporal snapshots, as detailed in Section 5.1. Each snapshot functions as a distinct task for continual learning, with subsequent snapshots introducing new facts and potentially new entity links.

**Incremental Training Procedure**: Training commences on the model $\mathcal{M}$ using $D_1^{train}$, with $D_1^{val}$ employed for hyper-parameter tuning. At each time step $t$, the model $\mathcal{M}_t$ with parameter set $\theta_t$ is initialized with parameters from the preceding time step $\theta_{t-1}$. Subsequently, $\mathcal{M}_t$'s parameters are updated by training on $D_t^{train}$. As illustrated in Figure 1, in the extrapolation setup, the training, validation, and testing datasets are constructed to ensure no temporal overlap. Specifically, event timestamps in the training data precede those in both the validation and test datasets. This design ensures the model is trained on historical data and evaluated on future quadruples with unseen timestamps. However, for incremental learning, this approach may omit data segments containing pivotal information. Thus, at training step $t$, we maintain two checkpoints: one post-training on $D_t^{train}$ for evaluation and another post-training on both the validation and test sets for a few epochs before proceeding to $D_{t+1}^{train}$. This incremental training can be a straightforward fine-tuning or can incorporate advanced incremental training strategies. For our base model, we employ TiTer Sun et al. (2021), which is the state-of-the-art in inductive temporal knowledge graph completion. TiTer leverages reinforcement learning to facilitate path-based reasoning.

**Comparison**. Besides the naive fine-tuning (FT) approach, we implement several variations of the base model, each enhanced with different Continual Learning (CL) strategies. These strategies include a regularization-based method known as Elastic Weight Consolidation (EWC) Kirkpatrick

Table 2: Total Link prediction performance comparison. Hit@10, Hit@3, Hit@1 and MRR reported for different models incrementally trained using two benchmarks: ICEWS18 and ICEWS14. Performance is evaluated at the final training time step over the last test dataset(Current) and across all prior test datasets (Average). FT: Fine-Tuning, EWC: Elastic Weight Consolidation, ER: Experience Replay. The first line is the model trained only on the first snapshot.

| | ICEWS18 | | | | | | | | ICEWS14 | | | | | | | |
| | Current | | | | Average | | | | Current | | | | Average | | | |
| Model | H@10 | H@3 | H@1 | MRR | H@10 | H@3 | H@1 | MRR | H@10 | H@3 | H@1 | MRR | H@10 | H@3 | H@1 | MRR |
|---|---|---|---|---|---|---|---|---|---|---|---|---|---|---|---|---|
| Titer | 0.443 | 0.317 | 0.206 | 0.287 | 0.451 | 0.336 | 0.224 | 0.303 | 0.571 | 0.466 | 0.364 | 0.436 | 0.555 | 0.428 | 0.286 | 0.380 |
| + FT | 0.444 | 0.324 | 0.214 | 0.295 | 0.464 | 0.350 | 0.229 | 0.312 | 0.582 | 0.494 | 0.393 | 0.464 | 0.572 | 0.456 | 0.326 | 0.413 |
| + ER | 0.487 | 0.353 | 0.234 | 0.319 | 0.470 | 0.349 | 0.231 | 0.314 | 0.593 | 0.477 | 0.379 | 0.452 | 0.572 | 0.443 | 0.304 | 0.396 |
| + EWC | 0.484 | 0.364 | 0.239 | 0.326 | 0.476 | 0.357 | 0.234 | 0.319 | 0.588 | 0.486 | 0.395 | 0.463 | 0.574 | 0.451 | 0.311 | 0.403 |
| + Ours | **0.496** | **0.368** | **0.242** | **0.330** | **0.489** | **0.362** | **0.236** | **0.323** | **0.610** | **0.517** | **0.407** | **0.482** | **0.581** | **0.469** | **0.334** | **0.421** |

et al. (2017), and an Experience Replay method (ER) Rolnick et al. (2019) that randomly selects a subset of data points to be incorporated into the training data for the subsequent step.

**Evaluation Metrics**. Similar to other KG completion studies, Liang et al. (2022), we evaluate the models using Mean Reciprocated Rank (MRR) and Hit@k metrics for $k = 1, 3, 10$. Following Mirtaheri et al. (2023), and in order to assess the ability of the model in alleviating the forgetting problem, at the current snapshot $t$, we report the average model performance over all the test data from previous snapshots, as well as the current test data.

**Implementation Detail**. We adopted the hyperparameters of the base model, Titer, as outlined in their paper and the accompanying code repository. For the enhancement layer and weighted sampling, we performed an extensive grid search. The search space encompassed: $\lambda \in [0.3, 0.5, 0.7]$, $\mu \in [0.1, 0.3, 0.5]$, the number of similar entities within $[10, 15, 20, 25]$, and the weighted sampling fraction $\alpha \in [0, 0.1, 0.2, 0.5, 0.8, 1]$. Model selection was based on achieving the highest average incremental performance on the validation set.

## 5.3 OVERALL PERFORMANCE (Q1)

Our evaluation primarily focuses on the link prediction performance across different models, incrementally trained using two benchmarks: ICEWS18 and ICEWS14. The results are summarized in Table 2. When trained only on the initial snapshot, the base model, Titer, demonstrated limited performance for both the latest snapshot and the average across all test datasets, as evident from the table's first row. This highlights the model's difficulty in effectively transferring knowledge. For other model variations, the table showcases the performance of the final incrementally trained model, $\mathcal{M}_T$, on the last snapshot and its average performance across all test datasets. While Fine-Tuning (FT) offered a modest improvement over the base model, both Experience Replay (ER) and Elastic Weight Consolidation (EWC) yielded more substantial gains. Incorporating our proposed framework into the base model led to superior performance across all configurations, with a more significant gain in MRR and Hit@10. Notably, our approach resulted in a 10% improvement in MRR for ICEWS14 and a 15% enhancement for ICEWS18, underscoring the effectiveness of our method in improving link prediction, especially within an incremental training framework.

## 5.4 INDUCTIVE LINK PREDICTION (Q2)

The inductive link prediction performance of various models, incrementally trained using the benchmarks ICEWS18 and ICEWS14, is presented in Table 3. This evaluation assesses the model's ability to predict links involving entities that were not observed during training. In our incremental framework, an entity $s$ at training step $t$ is considered unseen if it hasn't appeared in graph in previous steps and also the current training data, i.e., $s \notin \bigcup_{j=1}^{t-1} \mathcal{Q}_j \cup D_t^{train}$.

We present the inductive link prediction results for the model $\mathcal{M}_1$, which was trained on the initial snapshot, evaluated against the test dataset $D_1^{test}$ corresponding to that snapshot, denoted as "First" in the table. This snapshot is especially significant as it is the most extensive and contains the maximum

Table 3: Inductive Link prediction performance comparison. Hit@10, Hit@3, Hit@1 and MRR reported for different models incrementally trained using ICEWS18 and ICEWS14. Performance is evaluated at the first inductive test set (First) and the union of all inductive test sets at the last training time step (Average). FT: finetuning, EWC: Elastic Weight Consolidation, ER: Experience Replay.

| Model | ICEWS18 | | | | | | | | ICEWS14 | | | | | | | |
| | First | | | | Average | | | | First | | | | Average | | | |
| | H@10 | H@3 | H@1 | MRR | H@10 | H@3 | H@1 | MRR | H@10 | H@3 | H@1 | MRR | H@10 | H@3 | H@1 | MRR |
|---|---|---|---|---|---|---|---|---|---|---|---|---|---|---|---|---|
| Titer + FT | 0.448 | 0.334 | 0.219 | 0.298 | 0.091 | **0.073** | 0.048 | 0.064 | 0.547 | 0.421 | 0.285 | 0.376 | 0.146 | 0.126 | 0.097 | 0.115 |
| + ER | | | | | 0.090 | 0.069 | 0.045 | 0.061 | | | | | **0.147** | 0.118 | 0.081 | 0.103 |
| + EWC | | | | | 0.089 | 0.070 | 0.044 | 0.060 | | | | | 0.144 | 0.109 | 0.074 | 0.098 |
| + Ours | **0.458** | **0.342** | **0.223** | **0.305** | **0.092** | 0.071 | **0.054** | **0.066** | **0.564** | **0.439** | **0.295** | **0.388** | 0.146 | **0.128** | **0.098** | **0.115** |

Table 4: Ablation study on different model components. Compares Hit@10, Hit@3, Hit@1 and MRR for the base model, Titer, augmented with Fine-Tuning (FT) against our framework with isolated components (Weighted Sampling and Enhancement Layer) and their combined effect.

| Model | ICEWS18 | | | | | | | | ICEWS14 | | | | | | | |
| | Current | | | | Average | | | | Current | | | | Average | | | |
| | H@10 | H@3 | H@1 | MRR | H@10 | H@3 | H@1 | MRR | H@10 | H@3 | H@1 | MRR | H@10 | H@3 | H@1 | MRR |
|---|---|---|---|---|---|---|---|---|---|---|---|---|---|---|---|---|
| Titer + FT | 0.444 | 0.324 | 0.214 | 0.295 | 0.464 | 0.350 | 0.229 | 0.312 | 0.582 | 0.494 | 0.393 | 0.464 | 0.572 | 0.456 | 0.326 | 0.413 |
| + Ours (Weighted Sampling) | 0.482 | 0.366 | 0.237 | 0.325 | 0.477 | 0.359 | 0.236 | 0.320 | 0.621 | 0.489 | 0.393 | 0.467 | 0.579 | 0.459 | 0.321 | 0.412 |
| + Ours (Enhancement Layer) | 0.490 | 0.367 | 0.246 | 0.330 | 0.484 | 0.365 | 0.239 | 0.325 | 0.579 | 0.483 | 0.393 | 0.459 | 0.583 | 0.468 | 0.343 | 0.427 |
| + Ours (Full) | 0.496 | 0.368 | 0.242 | 0.330 | 0.489 | 0.362 | 0.236 | 0.323 | 0.610 | 0.517 | 0.407 | 0.482 | 0.581 | 0.469 | 0.334 | 0.421 |

number of quadruples. Furthermore, we present the inductive link prediction performance of the final model, $\mathcal{M}_T$, evaluated across all test sets from previous snapshots.

For the first snapshot, the results for FT, EWC, and ER are identical since, during the initial training step, they all utilize the same data and model architecture. Consequently, these results are omitted from the table. In contrast, our framework, which incorporates weighted sampling and the enhancement layer, was integrated into the TiTer architecture right from the initial training step.

When integrated with the base model, Titer, our method yielded substantial improvements across all metrics for the first snapshot. It also surpassed other baselines in the incremental task for ICEWS18, particularly in MRR and Hit@1. However, the task of incremental inductive link prediction proved to be more challenging, evident from the significant performance drop across all models. This complexity likely stems from the emergence of links in certain snapshots that are inherently more difficult to predict, leading to a pronounced decline in the performance metrics for all models.

## 5.5 ABLATION STUDY (Q3)

To elucidate the individual contributions of the components within our proposed framework, we conducted an ablation study. The results are presented in Table 4. We evaluated the base model, Titer, augmented with Fine-Tuning (FT) as a baseline. Subsequently, we assessed the performance of our framework with individual components: Weighted Sampling, Enhancement Layer, and the full combination of both. For the full model the hyper parameters are selected that achieve the best overall link prediction performance. For the isolated components, the hyper parameters were kept identical to the parts of the full model.

The results indicate that each component of our framework contributes to the overall performance improvement. Specifically, the Enhancement Layer and Weighted Sampling individually boost the model's performance, with the full combination of both components achieving the best results across all metrics. This ablation study underscores the synergistic effect of our proposed components in enhancing link prediction performance, especially in the context of incremental training.

To further analyze the contributions of each component in incremental learning, we assess how effectively each model mitigates catastrophic forgetting. Specifically, at time $t$, we compute the average

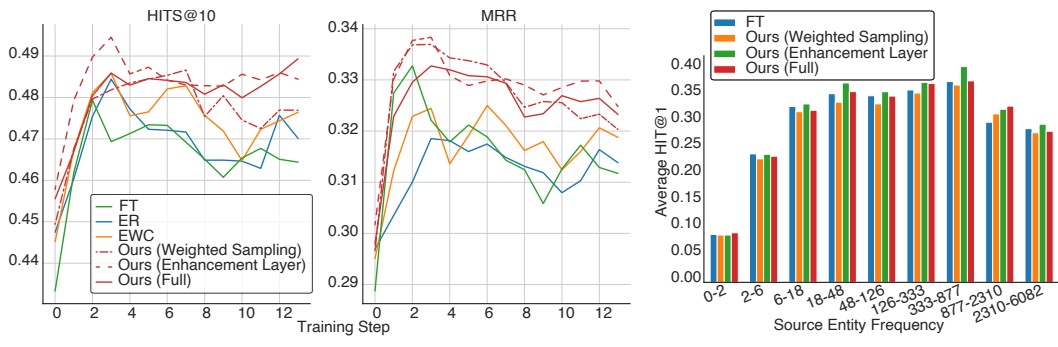

(a) Continual Learning Performance on ICEWS18

(b) Hit@1 Performance over the union of test datasets grouped by source entity frequency

Figure 2: Analysis of model components for catastrophic forgetting and long-tail entities.

performance of $\mathcal{M}_t$ over the current test set and all preceding ones, i.e., $D_1^{test}, D_2^{test}, \ldots, D_t^{test}$. We define the performance at time $t$ as $P_t = \frac{1}{t} \sum_{j=1}^{t} p_{t,j}$, where $p_{t,j}$ represents the performance of $\mathcal{M}_t$ on $D_j^{test}$, measured by metrics such as MRR or Hit@10. As illustrated in Figure 2a, our model's components, both individually and in combination, excel over other baselines in mitigating catastrophic forgetting, evident from the reduced performance decline over time. Notably, the enhancement layer demonstrates pronounced efficacy in this regard. While EWC surpasses other baselines, Experience Replay (ER) and Fine-Tuning (FT) exhibit marginal performance variations.

Furthermore, we examine the efficacy of each component in catering to long-tail entities. Figure 2b delineates the performance of each model on the union of test sets $D_1^{test}, \ldots, D_T^{test}$. Test set quadruples are aggregated based on the incremental frequency of their source entity. This incremental frequency, distinct from the overall frequency, corresponds to the entity's occurrence rate in the graph when the specific query quadruple was observed. The findings reveal that for entities with frequencies below 18, none of the models show a significant difference in the Hit@1 score. However, for frequencies exceeding 18, our model consistently surpasses the fine-tuned variant. The enhancement layer, in particular, significantly outperforms the fine-tuned model, especially for frequencies within the [18-48] range and beyond.

## 6 CONCLUSION

In this work, we developed an incremental training framework for Temporal Knowledge Graphs (TKGs) by incorporating a model-agnostic enhancement layer and a weighted sampling strategy. When augmented to GNN-based TKG completion methods, our approach yielded improvements in overall link prediction performance. Compared to models trained without incremental adaptation or other continual learning baselines, our method demonstrated better performance in mitigating catastrophic forgetting, with the enhancement layer being particularly influential in this aspect. Our framework also exhibited improved results in inductive link prediction and effectively addressed long-tail entities, especially those with moderately populated neighborhoods. While our approach marks a step forward in TKG completion, there remain areas for improvement, particularly for entities with extremely sparse neighborhoods. Future research directions could include the integration of richer features and continuous extraction of facts from LLMs to further refine TKG representations.

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

## APPENDIX

## A COMPLEXITY ANALYSIS

**Proposition 1.** STORAGE *complexity of the enhancement layer is* $\mathcal{O}(|\mathcal{B}|nd)$ *where* $\mathcal{B}$ *is the batch and* $n$ *is the maximum number of similar entities.*

**Proposition 2.** TIME *complexity of the enhancement layer is* $\mathcal{O}(|\mathcal{B}|nd)$ *for one forward pass, where* $\mathcal{B}$ *is the batch,* $n$ *is the maximum number of similar entities per entity and* $d$ *is the entity embedding dimensions.*

At each forward pass, for each quadruple in the batch, the algorithm has access to $n$ other similar entities and retrieves and keep their embeddings in the memory. Each forward pass of the enhancement layer involves elementwise multiplication of the retrieved entities with the exponential coefficients and then a temporal averaging which takes $\mathcal{O}(|\mathcal{B}|nd)$. Table 5 reports the memory and time complexity of each step of our approach both theoretically and in dataset ICEWS14.

| Methodology | Time Complexity | Memory Complexity | Per Epoch Runtime | Total Runtime |
|---|---|---|---|---|
| Naive Titer | - | - | ~9s | ~90m |
| Enhancement Layer | $\mathcal{O}(bnd)$ | $\mathcal{O}(bnd)$ | ~12s | ~125m |
| Weighted Sampling | $\mathcal{O}(n)$ | $\mathcal{O}(\|\mathcal{E}\|)$ | ~9s | ~110m |
| Full | $\mathcal{O}(bnd)$ | $\mathcal{O}(\|\mathcal{E}\| + bnd)$ | ~12s | ~125m |

Table 5: Comparative Analysis of Time and Memory Complexities for Various Models: Includes the enhancement layer, weighted sampling, and the full model. Also reports per epoch and total runtime for the naive Titer and Titer integrated with our models, applied to the ICEWS14 dataset

# B ADDITIONAL ENHANCEMENT LAYER IMPLEMENTATION

In addition to Titer, we have incorporated the enhancement layer into another state-of-the-art method TANGO Han et al. (2021b). We are providing the comparison between the naive fine-tuning and the model incorporated with the enhancement layer. To concisely illustrate the integration of the enhancement layer in TANGO, which comprises two graph convolutional layers, we explored two approaches: (1) positioning the enhancement layer above the two convolutional layers, and (2) placing it between them. Our findings indicate that situating the enhancement layer atop the convolutional layers results in better performance. The results are provided in Table 6.

| Model | Current | | | | Average | | | |
|---|---|---|---|---|---|---|---|---|
| | H@10 | H@3 | H@1 | MRR | H@10 | H@3 | H@1 | MRR |
| TANGO | 0.519 | 0.355 | 0.232 | 0.317 | 0.534 | 0.395 | 0.270 | 0.359 |
| TANGO + Enhancement Layer | 0.720 | 0.555 | 0.379 | 0.496 | 0.715 | 0.557 | 0.386 | 0.498 |

Table 6: Performance Comparison of TANGO and TANGO with Enhancement Layer reported for ICEWS14 dataset.

