# OpenReview forum: "Enhancing Temporal Knowledge Graph Completion with Global Similarity and Weighted Sampling"
_ICLR.cc/2024/Conference — Submitted to ICLR 2024_

### Official Review · Reviewer_wF5W · 2023-10-27

**Soundness:** 3 good
**Presentation:** 3 good
**Contribution:** 3 good
**Rating:** 6
**Confidence:** 3

**Summary:**

In summary, the paper presents an incremental learning approach for TKG completion that incorporates techniques like weighted sampling and a model-agnostic enhancement layer to address challenges of handling unseen and sparsely connected entities in a growing knowledge graph over time. Evaluation on the constructed benchmarks demonstrates the effectiveness of the proposed framework. The paper

**Strengths:**

1. The paper tackles the relatively new problem of incremental learning for temporal knowledge graph completion, which has practical values in real-world applications, as most real-world knowledge bases must address as data arrives continuously over time.
2.  It proposes a novel model-agnostic enhancement layer that leverages global entity similarity, providing a creative approach beyond local neighborhood proximity used in existing methods.
3. Results demonstrate substantial quantitative gains over baselines, indicating the high technical quality of the proposed framework components.

**Weaknesses:**

1. Real-world deployment considerations are not discussed. For example, analyzing memory/compute needs and ability to handle streaming data.
2. While weighted sampling is an intuitive idea, the sampling function used could be further explored and justified. For example, analyzing alternative frequency-based formulas or evaluating sampling directly from a learned importance weighting.

**Questions:**

1. In Figure 2.a, it seems that the proposed method (Full) is outperformed by the proposed method (Enhancement Layer) on HITS@10, and the proposed method (Full) is outperformed by the proposed method (Weighted Sampling), any insights on that?

---

> ### Author Response · Authors · 2023-11-23
>
> We appreciate the reviewer's time and insightful feedback.
>
> >1. Real-world deployment considerations are not discussed. For example, analyzing memory/compute needs and ability to handle streaming data.
>
> That is a great point. We have now added a table that report the memory and time complexity of each step of our approach both theoretically and in dataset ICEWS14. The table is copied below for ease of access.
>
> **Notation**:  $b$, $n$ and $d$ represent the batch size, number of similar entities, and the entity embedding dimensions. $|\mathcal{Q}|$ and $|\mathcal{E}|$ represent the number of entities and quadruples in the dataset respectively.
>
> | Methodology | Time Complexity | Memory Complexity | Per Epoch Runtime | Total Runtime |
> |-----------|-----------|-----------|-----------|-----------|
> | Naive Titer | - | - | ~9s | ~90m |
> | Enhancement Layer | $\mathcal{O}(bnd)$ | $\mathcal{O}(bnd)$  | ~12s | ~125m|
> | Weighted Sampling | $\mathcal{O}(n)$ | $\mathcal{O}(\|\mathcal{E}\|)$ | ~9s | ~110m |
> | Full | $\mathcal{O}(bnd)$ | $\mathcal{O}(\|\mathcal{E}\| + bnd)$ | ~12s | ~125m|
>
> >While weighted sampling is an intuitive idea, the sampling function used could be further explored and justified. For example, analyzing alternative frequency-based formulas or evaluating sampling directly from a learned importance weighting.
>
> We thank the reviewer for this comment. We conducted experiments with various strategies, including mean, max, and min inverse frequencies of head and tail entities. During our hyperparameter selection process, the approach we ultimately chose was the one that yielded the best performance. In terms of exploring a learned importance weighting for each entity, we agree that this is a promising direction. We have mentioned this point in the future work section.
>
> >In Figure 2.a, it seems that the proposed method (Full) is outperformed by the proposed method (Enhancement Layer) on HITS@10, and the proposed method (Full) is outperformed by the proposed method (Weighted Sampling), any insights on that?
>
> This is a great question. Our findings suggest that the enhancement layer improves performance for edges involving low and medium-degree entities, which aligns with its intended function. However, for edges involving higher-degree entities, the full methodology outperforms the individual components.

---

### Official Review · Reviewer_eLoE · 2023-10-28

**Soundness:** 3 good
**Presentation:** 3 good
**Contribution:** 1 poor
**Rating:** 3
**Confidence:** 5

**Summary:**

This paper presents an incremental training framework for Temporal Knowledge Graph (TKG) completion, addressing the challenges of generalizing new knowledge and managing sparse connections. The framework combines a model-agnostic enhancement layer that leverages global entity similarity and a weighted sampling strategy to improve link prediction and handle long-tail entities.

**Strengths:**

Clear writing, making it easy to understand the author's ideas.

**Weaknesses:**

W1. The experimental dataset is very single. ICEWS14 and 18 are only different in years, and the homogeneity is serious. Moreover, the author reconstructed ICEWS without even giving a detailed data description of the new dataset.

W2. The comparison method is single. The author mentioned a lot of related work, but in the end only one method was selected as the basic model for the experiment. Such an experiment does not prove that the enhancement method proposed by the author is universal.

W3. The method is too simple. The global similarity and weighted sampling proposed in the paper are superficial training strategies. There is no in-depth discussion and unique insights into the problem, and there is a lack of breakthrough contributions.

**Questions:**

Q1. Have you tried experimenting on more TKGs, such as WIKIDATA and GDELT?

Q2. Have you tried more TKGC methods? For example, Cygnet, LCGE?

Q3. Since global similarity is used to enhance the representation of entities, why not just cancel the incremental training setting and train the representations of all entities together from the beginning?

Q4. Generally speaking, incremental training only reduces training costs and should not be more effective than training from scratch. If you want to use incremental training to enhance the effect of TKGC, is this starting point wrong?

---

> ### Author Response · Authors · 2023-11-23
> **Experimental results of a new base model and further clarification of datasets**
>
> We appreciate the reviewer's time and insightful feedback. The reviewer has thoughtfully highlighted important considerations regarding the datasets:
> >Q1. Have you tried experimenting on more TKGs, such as WIKIDATA and GDELT?
>
> While we agree that applying our approach to additional datasets would indeed be interesting, we were unfortunately unable to do so within the timeframe of this revision. However, it is important to note the inherent diversity of the two datasets currently included in our manuscript. Although ICEWS14 and ICEWS18 originate from the same ICEWS dataset, they are recognized as distinct benchmarks in the literature due to their differing distributions and densities. For example, ICEWS18 has 10 times more quadruples compared to ICEWS14 despite both of them covering a similar time duration within their respective years (see table below). The section 5.1 of our paper describes the dataset construction. We have now also revised this section to explain the dataset construction in more detail. The following table summarizes the statistics across these datasets:
>
> | Dataset | \# Entities | \# Relations | \# Snapshots | \# Quads ($G_1$) | Ave. Quads ($G_{i> 1}$)|
> |-----------|-----------|-----------|-----------|-----------|-----------|
> | ICEWS14 | 7128 | 230 | 33 | ~28k/3.7k/4k| ~1k/0.3k/0.3k |
> | ICEWS18 | 23039 | 230 | 16 | ~244k/45k/43k |  ~8K/1k/1k |
>
>
> > Q2. Have you tried more TKGC methods? For example, Cygnet, LCGE?
> We are now adding another state-of-the-art model  TANGO [1] to the paper. Preliminary results show that even the addition of our enhancement layer alone improves the performance. The results are provided in the following table:
>
> |  Methodology |  | Current | | |  | Average | | |
> |-----------|-----------|-----------|-----------|-----------|-----------|-----------|-----------|-----------|
> |  | **Hit@10** | **Hit@3** | **Hit@1**| **MRR** | **Hit@10** | **Hit@3** | **Hit@1**| **MRR** |
> TANGO |0.519 |0.355 |0.232 |0.317 |0.534 |0.395 |0.270 |0.359 |
> TANGO + Enhancement Layer |0.720 |0.555 |0.379 |0.496 |0.715 |0.557 |0.386 |0.498 |
>
> Further details about how the enhancement layer was incorporated to TANGO is provided in our paper.
>
>
> **References**
>
> [1] Han, Zhen, et al. "Learning neural ordinary equations for forecasting future links on temporal knowledge graphs." Proceedings of the 2021 Conference on empirical methods in natural language processing. 2021. [(Code)](https://github.com/TemporalKGTeam/TANGO)

---

> > ### Author Response · Authors · 2023-11-23
> > **Response to Q3 and Q4**
> >
> > >Q3. Since global similarity is used to enhance the representation of entities, why not just cancel the incremental training setting and train the representations of all entities together from the beginning?
> >
> > To clarify, global similarity in our context, refers to identifying similar entities that may not be directly connected. It goes beyond immediate local proximity (e.g., one-hop neighborhoods) but does not imply full access to the entire graph structure from the beginning. Our data is received in a streaming fashion, and at each timestep, we compute similar entities based on the graph structure observed up to that point. The incremental training setting is essential because it allows our model to adapt to new information as it becomes available, reflecting the evolving nature of real-world data. Training all entity representations together from the beginning would require complete, upfront knowledge of the graph, which is not feasible in many practical TKG scenarios where data continuously evolves. Therefore, the incremental training approach, combined with global similarity, offers a more realistic and effective solution for TKG completion tasks, especially in dynamic environments where new entities and relationships are constantly emerging.
> >
> > >Q4. Generally speaking, incremental training only reduces training costs and should not be more effective than training from scratch. If you want to use incremental training to enhance the effect of TKGC, is this starting point wrong?
> >
> > Effectiveness can be defined differently depending on the application. In some scenarios, a model's performance on recent data is crucial, while in others, consistent performance across all tasks holds greater importance. When focusing on recent tasks, incremental training can potentially outperform a model trained from scratch, as it may be more biased to the latest data. Conversely, for overall performance across all tasks, a model trained on the entire dataset outperforms a fine-tuned model. However, this approach can be impractically inefficient for real-world applications, especially when dealing with large, continuously evolving datasets. Our goal is to strike a balance between maintaining high performance on recent tasks and overall robustness across all tasks, while also ensuring reasonable computational efficiency. Furthermore, beyond mere incremental fine-tuning, the overall performance can be enhanced through techniques such as Elastic Weight Consolidation (EWC), Experience Replay (ER), and our proposed enhancement layer.

---

### Official Review · Reviewer_Pdc3 · 2023-10-31

**Soundness:** 2 fair
**Presentation:** 2 fair
**Contribution:** 2 fair
**Rating:** 5
**Confidence:** 3

**Summary:**

This paper presents an incremental training framework for temporal knowledge graphs by incorporating a model-agnostic enhancement layer and a weighted sampling strategy. The authors conduct extensive experiments on the two popular datasets and the results show the effectiveness of the proposed method. The target problem is interesting.

**Strengths:**

1. The authors conduct extensive experiments on the two popular datasets and the results show the effectiveness of the proposed method.
2. The paper is well written and the target problem is interesting.

**Weaknesses:**

1. The motivation is not well established.  It seems that the authors combine the two methods (global similarity and weighted sampling).
2. In Table 1, many various recent works should be considered and discussed:
Xu et al., 2023. Temporal knowledge graph reasoning with historical contrastive learning.
Zhang et al., 2023. Learning Long- and Short-term Representations for Temporal Knowledge Graph Reasoning.
Zhu et al., 2021. Learning from History: Modeling Temporal Knowledge Graphs with Sequential Copy-Generation Networks.

**Questions:**

1. The motivation is not well established.  It seems that the authors combine the two methods (global similarity and weighted sampling).
2. In Table 1, many various recent works should be considered and discussed:
Xu et al., 2023. Temporal knowledge graph reasoning with historical contrastive learning.
Zhang et al., 2023. Learning Long- and Short-term Representations for Temporal Knowledge Graph Reasoning.
Zhu et al., 2021. Learning from History: Modeling Temporal Knowledge Graphs with Sequential Copy-Generation Networks.

---

> ### Author Response · Authors · 2023-11-23
>
> We thank the reviewer for their time and their valuable feedback.
>
> > 1. The motivation is not well established. It seems that the authors combine the two methods (global similarity and weighted sampling).
>
> We have modified the paper to better explain the motivation for our approach. Briefly, the motivation for integrating these two methods stems from their complementary strengths in addressing the challenges of Temporal Knowledge Graph (TKG) completion, including sparse or non-existent local connections and training biases resulting from unbalanced distribution. The enhancement layer addresses the issue of sparse or non-existent local connections for infrequent (long-tail) entities, an area where traditional TKG completion methods, which primarily relying on local neighborhood proximity for entity representation, fall short. On the other hand, weighted sampling ensures that long-tail entities are not overlooked by the model during training. Our ablation study, detailed in Section X of the paper, demonstrates how each component contributes to addressing these challenges. Figure 2.a illustrates the effectiveness of the enhancement layer for continual learning. As depicted in Figure 2.b, it also contributes to improving performance for entities with mid-degree nodes. The full method enhances hit@1 for low-degree nodes.
>
> >2. In Table 1, many various recent works should be considered and discussed: Xu et al., 2023. Temporal knowledge graph reasoning with historical contrastive learning. Zhang et al., 2023. Learning Long- and Short-term Representations for Temporal Knowledge Graph Reasoning. Zhu et al., 2021. Learning from History: Modeling Temporal Knowledge Graphs with Sequential Copy-Generation Networks.
>
> We appreciate the reviewer's suggestion regarding the related work. To clarify, our experiments were designed to demonstrate the effectiveness of our framework and its ability to be used on top of any state-of-the-art TKG model. In Table 1, we compare the performance of a model before and after the application of our framework.  We have now added another state-of-the-art model  TANGO [1] to the paper. Preliminary results show that even the addition of our enhancement layer alone improves the performance.
>
> The results are provided in the following table:
>
> |  Methodology |  | Current | | |  | Average | | |
> |-----------|-----------|-----------|-----------|-----------|-----------|-----------|-----------|-----------|
> |  | **Hit@10** | **Hit@3** | **Hit@1** | **MRR** | **Hit@10** | **Hit@3** | **Hit@1** | **MRR** |
> TANGO |0.519 |0.355 |0.232 |0.317 |0.534 |0.395 |0.270 |0.359 |
> TANGO + Enhancement Layer |0.720 |0.555 |0.379 |0.496 |0.715 |0.557 |0.386 |0.498 |
>
>
>
> **References**
>
> [1] Han, Zhen, et al. "Learning neural ordinary equations for forecasting future links on temporal knowledge graphs." Proceedings of the 2021 Conference on empirical methods in natural language processing. 2021. [(Code)](https://github.com/TemporalKGTeam/TANGO)

---

### Official Review · Reviewer_1RrB · 2023-11-03

**Soundness:** 3 good
**Presentation:** 3 good
**Contribution:** 3 good
**Rating:** 6
**Confidence:** 4

**Summary:**

This paper is about the temporal knowledge graph completion problem and its challenges. Models proposed to solve this problem need to take into consideration the requirement for generalization and assimilation to new knowledge, and the sparseness or connection between newly introduced entities. To overcome these challenges, the authors propose an incremental training framework specifically designed for temporal knowledge graphs, a unique enhancement layer that can be integrated with various GNN-based temporal knowledge graph completion methods, and a weighted sampling strategy during the training process which emphasizes the connections of infrequent entities. The experimental setup contains two versions of ICEWS datasets, link prediction tasks (overall, inductive), hit@k and MRR evaluation metrics, and the comparison methods that include the baseline model (Titer) with three variations (FT, ER, EWC) and three proposed variations (weighted sampling, enhancement layer, full). The results show the overall improvement in the model when utilizing the proposed enhancement layer and the proposed weighted sampling.

**Strengths:**

1) The paper is about an interesting topic (TKG completion task). The authors highlight the challenges in this topic and provide with proposed solutions that can be adapted in existing TKG completion models.
2) The results show the performance improvement when adding the proposed layer and sampling.
3) The paper is well-written and well-structured. The authors have done a great job to describe the area of research, its limitations, provide the related works, give solutions for each limitation and support the proposed methodology with experiments around three research questions.

**Weaknesses:**

1) The proposed methodology is incremental. The proposed layer and sampling are extensions to an existing model, Titer.
2) It would be useful if the authors could add other state of the art models (from their related work) to the comparison models. The experiments would be more convincing and would make a stronger point if the authors could show how the proposed methodology can be added in other models too, and to be able to see the performance of other models with the proposed extensions.
3) Another useful aspect on the proposed methodology is to add the run times and the time complexities when the layer and the sampling strategy are added.

**Questions:**

The authors can respond to my 2) and 3) comments in the weaknesses section.

Minor comment: In section 5.2, 3rd sentence, the citation was not added/compiled properly on LaTeX.

**Details Of Ethics Concerns:**

No concerns.

---

> ### Author Response · Authors · 2023-11-23
> **Added experimental results of a new base model and runtime analysis**
>
> > 2. It would be useful if the authors could add other state of the art models (from their related work) to the comparison models. The experiments would be more convincing and would make a stronger point if the authors could show how the proposed methodology can be added in other models too, and to be able to see the performance of other models with the proposed extensions.
>
> We thank the reviewer for their suggestion. We are now adding another state-of-the-art model  TANGO [1] to the paper. Preliminary results show that even the addition of our enhancement layer alone improves the performance. The results are provided in the following table:
>
> |  Methodology |  | Current | | |  | Average | | |
> |-----------|-----------|-----------|-----------|-----------|-----------|-----------|-----------|-----------|
> |  | **Hit@10** | **Hit@3** | **Hit@1**| **MRR** | **Hit@10** | **Hit@3** | **Hit@1**| **MRR** |
> TANGO |0.519 |0.355 |0.232 |0.317 |0.534 |0.395 |0.270 |0.359 |
> TANGO + Enhancement Layer |0.720 |0.555 |0.379 |0.496 |0.715 |0.557 |0.386 |0.498 |
>
> To concisely illustrate the integration of the enhancement layer in TANGO, which comprises two graph convolutional layers, we explored two approaches: (1) positioning the enhancement layer above the two convolutional layers, and (2) placing it between them. Our findings indicate that situating the enhancement layer atop the convolutional layers results in superior performance. Further details about this implementation have been elaborated in our paper.
>
>
> > 3. Another useful aspect on the proposed methodology is to add the run times and the time complexities when the layer and the sampling strategy are added.
>
> This is a great suggestion. We have added a table that reports the runtime and memory complexity of different variations of our model (weighted sampling, enhancement layer, full) in comparison with the naive Titer, as well as the actual per-epoch and total runtime of our method on the ICEWS14 dataset. The table is copied below for ease of access.
>
> **Notation**:  $b$, $n$ and $d$ represent the batch size, number of similar entities, and the entity embedding dimensions. $|\mathcal{Q}|$ and $|\mathcal{E}|$ represent the number of entities and quadruples in the dataset respectively.
>
> | Methodology | Time Complexity | Memory Complexity | Per Epoch Runtime | Total Runtime |
> |-----------|-----------|-----------|-----------|-----------|
> | Naive Titer | - | - | ~9s | ~90m |
> | Enhancement Layer | $\mathcal{O}(bnd)$ | $\mathcal{O}(bnd)$  | ~12s | ~125m|
> | Weighted Sampling | $\mathcal{O}(n)$ | $\mathcal{O}(\|\mathcal{E}\|)$ | ~9s | ~110m |
> | Full | $\mathcal{O}(bnd)$ | $\mathcal{O}(\|\mathcal{E}\| + bnd)$ | ~12s | ~125m|
>
> > Minor comment: In section 5.2, 3rd sentence, the citation was not added/compiled properly on LaTeX.
>
> We thank the reviewer for their comment. We have edited the paper and revised the issue.
>
> **References**
>
> [1] Han, Zhen, et al. "Learning neural ordinary equations for forecasting future links on temporal knowledge graphs." Proceedings of the 2021 conference on empirical methods in natural language processing. 2021. [(Code)](https://github.com/TemporalKGTeam/TANGO)

---

### Meta-Review · Area_Chair_P2Yx · 2023-12-23

**Metareview:**

The paper presents an incremental training framework for Temporal Knowledge Graph (TKG) completion, introducing a model-agnostic enhancement layer and a weighted sampling strategy. This approach addresses the challenges of generalizing new knowledge and managing sparse connections in TKGs. While the paper is well-written, clearly articulating the proposed solutions, and demonstrates performance improvement in experiments, it has several limitations. The methodology primarily extends an existing model, and the experimental setup is limited in scope, heavily relying on ICEWS datasets without detailed data descriptions or a broad range of comparison methods. Reviewers suggest that the paper lacks a well-established motivation and does not sufficiently differentiate from or compare with various recent works in the field.   Overall, while the paper contains practical value in real-world applications and its creative approach to leveraging global entity similarity for TKG completion, it is unfortunately not strong enough to justify acceptance in comparison with other submissions.

**Justification For Why Not Higher Score:**

The novelty of the proposed approach is not sufficient, and the comparison with relevant recent works is limited.

**Justification For Why Not Lower Score:**

The paper is clearly written and demonstrates the performance improvement well in experiments.

---

### Decision · Program_Chairs · 2024-01-16

Reject